# Dynamic Time Lag Regression: Predicting What & When

**Mandar Chandorkar**
Centrum Wiskunde en Informatica
Amsterdam 1098XG

**Cyril Furtlehner**
INRIA-Saclay

**Bala Poduval**
University of New Hampshire
Durham, NH 03824

**Enrico Camporeale**
CIRES, University of Colorado
Boulder, CO

**Michèle Sebag**
CNRS − Univ. Paris-Saclay

## Abstract

This paper tackles a new regression problem, called *Dynamic Time-Lag Regression* (DTLR), where a cause signal drives an effect signal with an unknown time delay. The motivating application, pertaining to space weather modelling, aims to predict the near-Earth solar wind speed based on estimates of the Sun's coronal magnetic field. DTLR differs from mainstream regression and from sequence-to-sequence learning in two respects: firstly, no ground truth (e.g., pairs of associated subsequences) is available; secondly, the cause signal contains much information irrelevant to the effect signal (the solar magnetic field governs the solar wind propagation in the heliosphere, of which the Earth's magnetosphere is but a minuscule region).

A Bayesian approach is presented to tackle the specifics of the DTLR problem, with theoretical justifications based on linear stability analysis. A proof of concept on synthetic problems is presented. Finally, the empirical results on the solar wind modelling task improve on the state of the art in solar wind forecasting.

## 1 Introduction

A significant body of work in machine learning concerns the modeling of spatio-temporal phenomena (Shi and Yeung, 2018; Rangapuram et al., 2018), including the causal analysis of time series Peters et al. (2017), with applications ranging from markets (Pennacchioli et al., 2014) to bioinformatics (Brouard et al., 2016) to climate (Nooteboom et al., 2018).

This paper focuses on the problem of modeling the temporal dependency between two spatio-temporal phenomena, where the latter one is *caused* by the former one (Granger, 1969; Runge, 2018) with a non-stationary time delay.

The motivating application domain is that of space weather. The sun, a perennial source of charged energetic particles, is at the origin of geomagnetic phenomena within the sun-earth system. Specifically, the sun ejects charged particles into the surrounding space in all directions and some of these particle clouds, a.k.a. *solar wind*, reach the Earth's vicinity. High speed solar wind is a major threat for the modern world, causing severe damages to e.g., satellites, telecommunication infrastructures, under sea pipelines, among others.[1]

A key prediction task thus is to forecast the speed of the solar wind in the vicinity of the Earth (Munteanu et al., 2013; Haaland et al., 2010; Reiss et al., 2019), sufficiently early to emit an alarm and be able to prevent the damage to the best possible extent. Formally the goal is to model the dependency between heliospheric observations (available at light speed), referred to as *cause series*, and the solar wind speed series recorded at the Lagrangian point $L_1$ (a point on the Sun-Earth line 1.5 million kilometers away from the Earth), referred to as *effect series*. The key difficulty is that the

---

[1]The adverse impact of space weather is estimated to cost 200 to 400 million USD per year, but can sporadically lead to much larger losses.

time lag between an input and its effect, the solar wind recorded at $L_1$, varies from circa 2 to 5 days depending on, among many factors, the initial direction of emitted particles and their energy. Would the lag be constant, the solar wind prediction problem would boil down to a mainstream regression problem. The challenge here is to predict, from the solar image $x(t)$ at time $t$ the value $y(t + \Delta t)$ of the solar wind speed reaching the earth at time $t + \Delta t$ where both the value $y(t + \Delta t)$ and the time lag $\Delta t$ depend on $x(t)$.

**Related work.**   Indeed, the modeling of dependencies among time series has been intensively tackled (see e.g., Zhou and Sornette (2006); Runge (2018)). When considering varying time lag, many approaches rely on dynamic time warping (DTW) (Sakoe and Chiba, 1978). For instance, DTW is used in Gaskell et al. (2015), taking a Bayesian approach to achieve the temporal alignment of both series under some restricting assumptions (considering slowly varying time lags and linear relationships between the cause and effect time series). More generally, the use of DTW in time series analysis relies on simplifying assumptions on the cause and effect series (same dimensionality and structure) and builds upon available cost matrices for the temporal alignment.
Also related is sequence-to-sequence learning (Sutskever et al., 2014), primarily aimed to machine translation. While Seq2Seq modelling relaxes some former assumptions (such as the fixed or comparable sizes of the source and target series), it still relies on the known segmentation of the source series into disjoint units (the sentences), each one being mapped into a large fixed-size vector using an LSTM; and this vector is exploited by another LSTM to extract the output sequence. Attention-based mechanisms Graves (2013); Bahdanau et al. (2015) alleviate the need to encode the full source sentence into a fixed-size vector, by learning the alignment and allowing the model to search for the parts of the source sentence relevant to predict a target part. More advanced attention mechanisms (Kim et al., 2017; Vaswani et al., 2017) refine the way the source information is leveraged to produce a target part. But to our best knowledge, the end-to-end learning of the sequence-to-sequence modelling relies on the segmentation of the source and target series, and the definition of associated pairs of segments (e.g. the sentences).

Our claim is that the regression problem of predicting both *what* the effect is and *when* the effect is observed, called *Dynamic TimeLag Regression* (DTLR), constitutes a new ML problem:
With respect to the modeling of dependencies among time series, it involves stochastic dependencies of arbitrary complexity; the relationship between the cause and the effect series can be non-linear (the *what* model). Furthermore, the time lag phenomenon (the *when* model) can be non smooth (as opposed to e.g. Zhou and Sornette (2006)).
With respect to sequence-to-sequence translation, a main difference is that the end-to-end training of the model cannot rely on pairs of associated units (the sentences), adversely affecting the alignment learning.
Lastly, and most importantly, in the considered DTLR problem, even if the cause series has high information content, only a small portion of it is relevant to the prediction of the effect series. On one hand, the cause series might be high dimensional (images) whereas the effect series is scalar; on the other hand, the cause series governs the solar wind speed in the whole heliosphere and not just in near-Earth space. In addition to avoiding typically one or two orders of magnitude expansion of an already large input signal dimension, inserting the time-lag inference explicitly in the model can also potentially improve its interpretability.

**Organization of the paper.**   The Bayesian approach proposed to tackle the specifics of the DTLR regression problem is described in section 2; the associated learning equations are discussed, followed by a stability analysis and a proof of consistency (section 3). The algorithm is detailed in section 4. The experimental setting used to validate the approach is presented in section 5; the empirical validation on toy problems and on the real-world problem are discussed in section 6

## 2   PROBABILISTIC DYNAMICALLY DELAYED REGRESSION

### 2.1   POSITION OF THE PROBLEM

Given two time series, the cause series $\mathbf{x}(t)$ ($\mathbf{x}(t) \in \mathcal{X} \subset \mathbb{R}^D$) and the observed effect series $y(t)$, the sought model consists of a mapping $f(.)$ which maps each input pattern $\mathbf{x}(t)$ to an output $y(\phi(t))$, and a mapping $g(.)$ which determines the time delay $\phi(t) - t$ between the input and output patterns:

$$y(\phi(t)) = f[\mathbf{x}(t)] \qquad (1)$$
$$\phi(t) = t + g[\mathbf{x}(t)] \qquad (2)$$

with

$$f : \mathcal{X} \to \mathbb{R}, \qquad \text{and} \qquad g : \mathcal{X} \to \mathbb{R}^+,$$

where $t \in \mathbb{R}^+$ represents the continuous temporal domain. The input signal $\mathbf{x}(t)$ is possibly high dimensional and contains the hidden cause to the effect $y(t) \in \mathbb{R}$; $y(t)$ is assumed to be scalar in the remainder of the paper. Function $g : \mathcal{X} \to \mathbb{R}^+$ represents the time delay between inputs and outputs. Vectors are written using bold fonts.

As said, Eqs 1-2 define a regression problem that differs from standard regression in two ways: Firstly, the time lag $g[\mathbf{x}(t)]$ is non-stationary as it depends on $\mathbf{x}(t)$. Secondly, $g[\mathbf{x}(t)]$ is unknown, i.e. it is not recorded explicitly in the training data.

**Assumption.** For the sake of the model identifiability and computational stability, the time warping function $\phi(t) = t + g[\mathbf{x}(t)]$ is assumed to be sufficiently regular w.r.t. $t$. Formally, $\phi(.)$ is assumed to be continuous[2].

## 2.2 PROBABILISTIC DYNAMIC TIME-LAG REGRESSION

For practical reasons, cause and effect series are sampled at constant rate, and thereafter noted $\mathbf{x}_t$ and $y_t$ with $t$ in $\mathcal{N}$. Accordingly, mapping $g$ maps $\mathbf{x}_t$ onto a finite set $\mathcal{T}$ of possible time lags, with $\mathcal{T} = \{\Delta t_{\min} \ldots, \Delta t_{\max}\}$ an integer interval defined from domain knowledge. The unavoidable error due to the discretization of the continuous time lag is mitigated along a probabilistic model, associating to each cause $\mathbf{x}$, a set of predictors $\hat{\mathbf{y}}(\mathbf{x}) = \{\hat{y}_i(\mathbf{x}), i \in \mathcal{T}\}$ and a probability distribution $\hat{p}(\mathbf{x})$ on $\mathcal{T}$ estimating the probability of delay of the effects of $\mathbf{x}$. Overall, the DTLR solution is sought as a probability distribution conditioned on cause $\mathbf{x}$, mixture of Gaussians[3] centered on the predictors $\hat{y}_i(\mathbf{x})$, where the mixture weights are defined from $\hat{p}(\mathbf{x})$. More formally, letting $\mathbf{y}_t$ denote the vector of random variables $\{y_{t+i}, i \in \mathcal{T}\}$:

$$P\big[\mathbf{y}_t | \mathbf{x}_t = \mathbf{x}\big] = \sum_{\{\tau_i \in \{0,1\}, i \in \mathcal{T}\}} \hat{p}\big(\tau_1, \ldots, \tau_{|\mathcal{T}|} | \mathbf{x}\big) \mathcal{N}\big(\hat{\mathbf{y}}(\mathbf{x}), \Sigma(\tau)\big) \qquad (3)$$

with $\Sigma = Diag(\sigma_i(\tau)^2)$ the diagonal matrix of variance parameters attached to each time-lag $i \in \mathcal{T}$. Two simplifications are made for the sake of the analysis. Firstly, the stochastic time lag is modelled as the vector $(\tau_i), i \in \mathcal{T}$ of binary latent variables, where $\tau_i$ indicates whether $\mathbf{x}_t$ drives $y_{t+i}$ ($\tau_i = 1$) or not ($\tau_i = 0$). The assumption that every cause has a single effect is modelled by imposing: [4]

$$\sum_{i \in \mathcal{T}} \tau_i = 1. \qquad (4)$$

From constraint (4), probability distribution $\hat{\mathbf{p}}(\mathbf{x})$ thus is sought as the vector $(\hat{p}_i(\mathbf{x}))$ for $i$ in $\mathcal{T}$, summing to 1, such that $\hat{p}_i(\mathbf{x})$ stands for the probability of the effect of $\mathbf{x}_t = \mathbf{x}$ to occur with delay $i$. The second simplifying assumption is that the variance $\sigma_i^2(\tau)$ of predictor $\hat{y}_i$ does not depend on $\mathbf{x}$, by setting:

$$\sigma_i(\tau)^{-2} = \Big(1 + \sum_j \alpha_{ij} \tau_j\Big) \sigma^{-2},$$

with $\sigma^2$ a default variance and $\alpha_{ij} \geq 0$ a matrix of non-negative real parameters. This particular formulation supports the tractable analysis of the posterior probability of $\tau_i$ (in supplementary

---

[2] For some authors (Zhou and Sornette, 2006),the monotonicity of $\phi(.)$ is additionally required and enforced using constraints:

$$\phi(t_1) \leq \phi(t_2), \forall t_1 \leq t_2.$$

This assumption is not retained as one may achieve a similar effect by using regularization based smoothness penalties.

[3]Note that pre-processing can be used if needed to map non-Gaussian data onto Gaussian data.

[4]Note however that the cause-effect correspondence might be many-to-one, with an effect depending on several causes.

material). The fact that $\mathbf{x}$ can influence $y_i$ through predictor $\hat{y}_i(\mathbf{x})$ even when $\tau_i = 0$ reflects an indirect influence due to the auto-correlation of the $y$ series. This influence comes with a higher variance, enforced by making $\alpha_{ij}$ a decreasing function of $|i - j|$. More generally, a large value of $\alpha_{ii}$ compared to $\alpha_{ij}$ for $i \neq j$ corresponds to a small auto-correlation time of the effect series.

## 2.3 LEARNING CRITERION

The joint distribution is classically learned by maximizing the log likelihood of the data, which can here be expressed in closed form. Let us denote respectively the dataset and parameters as $\{(\mathbf{x}, \mathbf{y})\}_{\text{data}}$ and $\theta = (\hat{\mathbf{y}}, \hat{\mathbf{p}}, \sigma, \alpha)$. From Eq. (3) the conditional probability $q_i(\mathbf{x}, \mathbf{y}) \stackrel{\text{def}}{=} P(\tau_i = 1 | \mathbf{x}, \mathbf{y})$ reads:

$$q_i(\mathbf{x}, \mathbf{y}) = \frac{1}{Z(\mathbf{x}, \mathbf{y}|\theta)} \hat{p}_i(\mathbf{x}) \exp\left(-\frac{1}{2\sigma^2} \sum_{j \in \mathcal{T}} \alpha_{ji}\big(y_j - \hat{y}_j(\mathbf{x})\big)^2 + \frac{1}{2} \sum_{j \in \mathcal{T}} \log(1 + \alpha_{ji})\right) \quad (5)$$

with normalization constant

$$Z(\mathbf{x}, \mathbf{y}|\theta) = \sum_{i \in \mathcal{T}} \hat{p}_i(\mathbf{x}) \exp\left(-\frac{1}{2\sigma^2} \sum_{j \in \mathcal{T}} \alpha_{ji}\big(y_j - \hat{y}_j(\mathbf{x})\big)^2 + \frac{1}{2} \sum_{j \in \mathcal{T}} \log(1 + \alpha_{ji})\right).$$

The log-likelihood then reads (intermediate calculations in supplementary material, appendix A):

$$\mathcal{L}[\{(\mathbf{x}, \mathbf{y})\}_{\text{data}}|\theta] = -|\mathcal{T}| \log(\sigma) - \mathbb{E}_{\text{data}}\left[\sum_{i \in \mathcal{T}} \frac{1}{2\sigma^2}\big(y_i - \hat{y}_i(\mathbf{x})\big)^2 - \log\big(Z(\mathbf{x}, \mathbf{y}|\theta)\big)\right] \quad (6)$$

where $\mathbb{E}_{\text{data}}$ denotes averaging over the dataset. For notational simplicity, the time index $t$ is omitted in the following and the empirical averaging on the data is noted $\mathbb{E}_{data}$. The hyper-parameters $\sigma$ and matrix $\alpha$ of the model are obtained by optimizing $\mathcal{L}$:

$$\frac{\sigma^2}{1 + \alpha_{ij}} = \frac{\mathbb{E}_{data}\left[\big(y_i - \hat{y}_i(\mathbf{x})\big)^2 q_j(\mathbf{x}, \mathbf{y})\right]}{\mathbb{E}_{data}\big[q_j(\mathbf{x}, \mathbf{y})\big]}, \quad (7)$$

In addition the optimal $\hat{\mathbf{y}}$ and $\hat{\mathbf{p}}$ reads:

$$\hat{y}_i(\mathbf{x}) = \frac{\mathbb{E}_{data}\left[y_i\big(1 + \sum_{j \in \mathcal{T}} \alpha_{ij} q_j(\mathbf{x}, \mathbf{y})\big)\Big|\mathbf{x}\right]}{\mathbb{E}_{data}\left[1 + \sum_{j \in \mathcal{T}} \alpha_{ij} q_j(\mathbf{x}, \mathbf{y})\Big|\mathbf{x}\right]} \quad (8)$$

$$\hat{p}_i(\mathbf{x}) = \mathbb{E}_{data}\left[q_i(\mathbf{x}, \mathbf{y})\Big|\mathbf{x}\right], \quad (9)$$

where the above conditional empirical averaging operates as an averaging over samples close to $\mathbf{x}$.

These are self-consistent equations, since $q_i(\mathbf{x}, \mathbf{y})$ depends on the parameters $\sigma^2$ and $\alpha_{ij}$, $\hat{\mathbf{y}}$ and $\hat{\mathbf{p}}$. The proposed algorithm detailed in section 4 implements the saddle point method defined from Eqs (7,5,8,9): alternatively, hyper-parameters $\sigma$ and $\alpha_{ij}$ are updated from Eq. (7) based on the current $\hat{y}_i$ and $\hat{p}_i$; and predictors $\hat{y}_i$ and mixture weights $\hat{p}_i$ are updated according to Eqs (8) and (9) respectively.

## 3 THEORETICAL ANALYSIS

The proposed DTLR approach is shown to be consistent and analyzed in the simple case where $\alpha$ is a diagonal matrix ($\alpha_{ij} = \alpha \delta_{ij}$).

### 3.1 LOSS FUNCTION AND RELATED OPTIMAL PREDICTOR

Let us assume that the hyper-parameters of the model have been identified together with predictors $\hat{y}_i(\mathbf{x})$ and weights $\hat{p}_i(\mathbf{x})$. These are leveraged to achieve the prediction of the effect series. For any given input $\mathbf{x}$, the sought eventual predictor is expressed as $(\hat{y}(\mathbf{x}), \hat{I}(\mathbf{x}))$ where $\hat{I}(\mathbf{x})$ is the predicted time lag and $\hat{y}(\mathbf{x})$ the predicted value. The associated $L_2$ loss is:

$$\mathcal{L}_2(\hat{y}, \hat{I}) = \mathbb{E}_{data}\left[\big(y_{\hat{I}(\mathbf{x})} - \hat{y}(\mathbf{x})\big)^2\right]. \quad (10)$$

Then it comes:

**Proposition 3.1.** *With same notations as in Eq. (3), with $\alpha_{ij} = \alpha\delta_{ij}$, $\alpha > 0$, the optimal composite predictor $(y^\star, I^\star)$ is given by*

$$y^\star(\mathbf{x}) = \hat{y}_{I^\star(\mathbf{x})}(\mathbf{x}) \qquad with \qquad I^\star(\mathbf{x}) = \arg\max_i \big(\hat{p}_i(\mathbf{x})\big),$$

**Proof.** In supplementary material, Appendix C. ∎

### 3.2 LINEAR STABILITY ANALYSIS

The saddle point (Eqs 7, 5, 8, 9) admits among others a degenerate solution, corresponding to $\hat{p}_i(\mathbf{x}) = 1/|\mathcal{T}|$, $\alpha_{ij} = 0$ for all pairs $(i, j)$, with $\sigma^2 = \sigma_0^2$. Informally the model converges toward this degenerate trivial solution when there is not enough information to build specialized predictors $\hat{y}_i$.

Let us denote $\Delta y_i^2(\mathbf{x}) = \big(y_i - \hat{y}_i(\mathbf{x})\big)^2$ the square error made by predictor $\hat{y}_i$ for $\mathbf{x}$, and

$$\sigma_0^2 = \frac{1}{|\mathcal{T}|}\mathbb{E}_{data}\Big(\sum_{i\in\mathcal{T}} \Delta y_i^2(\mathbf{x})\Big)$$

the average of MSE over the set of the predictors $\hat{y}_i$, $i \in \mathcal{T}$.

Let us investigate the conditions under which the degenerate solution may appear, by computing the Hessian of the log-likelihood and its eigenvalues. Under the simplifying assumption

$$\alpha_{ij} = \alpha\delta_{ij},$$

the model involves $2|\mathcal{T}|$ functional parameters $\hat{\mathbf{y}}$ and $\hat{\mathbf{p}}$ and two hyper-parameters $\alpha$ and $r = \sigma^2/\sigma_0^2$. After the computation of the Hessian (in supplementary material, Appendix B) the system involves three key statistical quantities, two global ones:

$$C_1[\mathbf{q}] = \frac{1}{\sigma_0^2}\mathbb{E}_{data}\Big(\sum_{i\in\mathcal{T}} q_i(\mathbf{x}, \mathbf{y})\Delta y_i^2(\mathbf{x})\Big), \tag{11}$$

$$C_2[\mathbf{q}] = \frac{1}{\sigma_0^4}\mathbb{E}_{data}\Big[\sum_{i\in\mathcal{T}} q_i(\mathbf{x}, \mathbf{y})\Big(\Delta y_i^2(\mathbf{x}) - \sum_{j\in\mathcal{T}} q_j(\mathbf{x}, \mathbf{y})\Delta y_j^2(\mathbf{x})\Big)^2\Big], \tag{12}$$

and a local $|\mathcal{T}|$-vector of components

$$u_i[\mathbf{x}, \mathbf{q}] = \frac{1}{\sigma_0^2}\mathbb{E}_{data}\Big[q_i(\mathbf{x}, \mathbf{y})\big(\Delta y_i^2(\mathbf{x}) - \sum_{j\in\mathcal{T}} q_j(\mathbf{x}, \mathbf{y})\Delta y_j^2(\mathbf{x})\big)\Big|\mathbf{x}\Big].$$

Up to a constant, $C_1$ represents the covariance between the latent variables $\{\tau_i\}$ and the normalized predictor errors. $C_1$ smaller than one indicates a positive correlation between the latent variables and small errors; the smaller the better. For the degenerate solution, i.e. $\mathbf{q} = \mathbf{q}_0$ uniform, $C_1[\mathbf{q}_0] = 1$ and $C_2[\mathbf{q}_0]$ represents the default variability among the prediction errors. $u_i[\mathbf{x}, \mathbf{q}]$ informally measures the quality of predictor $\hat{y}_i$ relatively to the other ones at $\mathbf{x}$. More precisely, a negative value of $u_i[\mathbf{x}, \mathbf{q}]$ indicates that $\hat{y}_i$ is doing better than average in the neighborhood of $\mathbf{x}$.

At a saddle point the parameters are given by:

$$\frac{\sigma^2}{\sigma_0^2} = \frac{|\mathcal{T}| - C_1[\mathbf{q}]}{|\mathcal{T}| - 1} \qquad and \qquad \alpha = \frac{|\mathcal{T}|}{|\mathcal{T}| - 1}\frac{1 - C_1[\mathbf{q}]}{C_1[\mathbf{q}]}.$$

The predictors $\hat{\mathbf{y}}$ are decoupled from the rest whenever they are centered, which we assume. So the analysis can focus on the other parameters.

**If $\hat{\mathbf{p}}$ is fixed**   a saddle point is stable iff

$$C_2[\mathbf{q}] < 2C_1^2[\mathbf{q}] + \mathcal{O}\Big(\frac{1}{|T|}\Big).$$

In particular, the degenerate solution is unstable if

$$C_2[\mathbf{q}_0] > 2\Big(1 - \frac{1}{|\mathcal{T}|}\Big).$$

Note that for $\Delta y_i(\mathbf{x})$ iid centered with variance $\sigma_0^2$ and relative kurtosis $\kappa$ (conditionally to $\mathbf{x}$) one has $C_2 = (2 + \kappa)(1 - 1/|T|)$. Therefore, whenever $\Delta y_i^2(\mathbf{x})$ fluctuates and the relative kurtosis is non-negative, the degenerate solution is unstable and will thus be avoided.

**If $\hat{p}$ is allowed to evolve**   (after Eq. (9)) the degenerate trivial solution becomes unstable as soon as $C_2[\mathbf{q}_0]$ is non-zero, due to the fact that the gradient points in the opposite direction to $\mathbf{u}(\mathbf{x})$ (with $d\hat{\mathbf{p}}(\mathbf{x}) \propto -C_2[\mathbf{q}_0]\mathbf{u}(\mathbf{x})$), thus rewarding the predictors with lowest errors by increasing their weights.

The system is then driven toward other solutions, among which the localized solutions of the form:

$$\hat{p}_i(\mathbf{x}) = \delta_{i,I(\mathbf{X})},$$

with an input dependent index $I(\mathbf{x}) \in \mathcal{T}$. As shown (in supplementary material, Appendix C) the maximum likelihood localized solution also minimizes the loss function (Eq. 10). The stability of such localized solutions and the existence of other (non-localized) solutions is left for further work.

## 4   THE DTLR ALGORITHM

The DTLR algorithm learns both regression models $\hat{\mathbf{y}}(\mathbf{x})$ and $\hat{\mathbf{p}}(\mathbf{x})$ from series $\mathbf{x}_t$ and $y_t$, using alternate optimization of the model parameters and the model hyper-parameters $\alpha$ and $\sigma^2$, after Eqs (7,5,8,9). The model search space is that of neural nets, parameterized by their weight vector $\theta$. The inner optimization loop updates $\theta$ using mini-batch based stochastic gradient descent. At the end of each epoch, after all minibatches have been considered, the outer optimization loop computes hyper-parameters $\alpha$ and $\sigma^2$ on the whole data.

**Initialization** of $\alpha$ and $\sigma$
$it \longleftarrow 0$ ;
**while** $it < max$ **do**
    **while** *epoch* **do**
        |   $\theta \longleftarrow Optimize(\mathcal{L}(\theta, \alpha, \sigma^2))$ ;
    **end**
    $\sigma^2 \longleftarrow \sigma_0^2 \frac{|T|-C_1[\mathbf{q}]}{|T|-1}$ ;
    $\alpha \longleftarrow \frac{|T|}{|T|-1} \frac{1-C_1[\mathbf{q}]}{C_1[\mathbf{q}]}$ ;
**end**
**Result:** Model parameters $\theta = \{\hat{\mathbf{y}}, \hat{\mathbf{p}}\}$, hyper-parameters $\alpha, \sigma^2$

**Algorithm 1:** DTLR algorithm

The algorithm code is available in supplementary material and will be made public after the reviewing period. The initialization of hyper-parameters $\alpha$ and $\sigma$ is settled using preliminary experiments (same setting for all considered problems: $\alpha \sim U(0.75, 2)$; $\sigma^2 \sim U(10^{-5}, 5)$).

The neural architecture implements predictors $\hat{\mathbf{y}}(\mathbf{x})$ and weights $\hat{\mathbf{p}}(\mathbf{x})$ on the top of a same feature extractor from input $\mathbf{x}$. In the experiments, the architecture of the feature extractor is a 2-hidden layer fully connected network. On the top of the feature extractor are the single layer $\hat{\mathbf{y}}$ and $\hat{\mathbf{p}}$ models, each with $|T|$ output neurons, with $|T|$ the size of the chosen domain for the time lag.

## 5   EXPERIMENTAL SETTING

The goal of experimental validation is threefold. A first issue regards the accuracy of DTLR, measured from the mean absolute error (MAE), root mean square error (RMSE) and Pearson correlation of the learned DTLR model $(y^\star(\mathbf{x}_t), I^\star(\mathbf{x}))$. DTLR is compared to the natural baseline defined as the regressor with constant time lag, $\hat{y}_{\bar{\Delta}}(\mathbf{x}_t)$, with $\bar{\Delta}$ being the average of all possible time lags in $\mathcal{T}$. The predictions of DTLR and the baseline are compared with the ground truth value of the effect series. However the predicted time lag $I^\star(\mathbf{x}_t)$ can only be assessed if the ground truth time-lag relationship is known. In order to do so, three synthetic problems of increasing difficulty are defined below.

Secondly, the stability and non-degeneracy of the learned model are assessed from the statistical quantities $\sigma_0$ and $C_1$ (section 3.2), compared to the degenerate solution $\hat{p}_i(\mathbf{x}) = 1/|\mathcal{T}|$. For $C_1 < 1$, the model accurately specializes the found predictors $\hat{p}_i$.

Lastly, and most importantly, DTLR is assessed on the solar wind prediction problem, and compared to the best state of the art in space weather.

**Synthetic Problems.** Four synthetic problems of increasing difficulty are generated using *Stochastic Langevin Dynamics*. In all problems, the cause signal $\mathbf{x}_t \in \mathbb{R}^{10}$ and the effect signal $y_t$ are generated as follows (with $\eta = 0.02, s^2 = 0.7$):

$$\mathbf{x}_{t+1} = (1 - \eta)\mathbf{x}_t + \mathcal{N}(0, s^2) \tag{13}$$

$$v_t = k||\mathbf{x}_t||^2 + c \tag{14}$$

$$y_{t+g(\mathbf{x}_t)} = f(v_t), \tag{15}$$

with time-lag mapping $g(\mathbf{x}_t)$ ranges in a time interval with width 20 (except for problem I where $|\mathcal{T}| = 15$). The complexity of the synthetic problems is governed by the amplitude and time-lag functions $f$ and $g$ (more in appendix, Table 2):

| Problem | $f(v_t)$ | $g(\mathbf{x}_t)$ | Other |
|---|---|---|---|
| **I** | $v_t$ | 5 | $k{=}10, c{=}0$ |
| **II** | $v_t$ | $100/v_m$ | $k{=}1, c{=}10$ |
| **III** | $\sqrt{v_t^2 + 2ad}$ | $(\sqrt{v_m^2 + 2ad} - v)/a$ | $k{=}5, a{=}5, d{=}1000, c{=}100$ |
| **IV** | $v_t$ | $g(\mathbf{x}_t){=}\exp(v_t)/(1{+}\exp(v_t/20))$ | $k{=}10, c{=}40$ |

**Solar Wind Speed Prediction.** The challenge of predicting solar wind speed from heliospheric data is due to the non-stationary propagation time of the solar plasma through the interplanetary medium. For the sake of a fair comparison with the best state of the art Reiss et al. (2019), the same experimental setting is used. The cause series $\mathbf{x}_t$ includes the solar magnetic (*flux tube expansion*, FTE: the rate at which the magnetic flux tubes expand between the photosphere and a reference height in the corona at 2.5 solar radii) and the coronal magnetic field strength estimates produced by the Current Sheet Source Surface model (Zhao and Hoeksema, 1995; Poduval and Zhao, 2014; Poduval, 2016), exploiting the hourly magnetogram data recorded by the *Global Oscillation Network Group* from 2008 to 2016. The effect series, the hourly solar wind data is available from the OMNI data base from the *Space Physics Data Facility* [5]. After domain knowledge, the time-lag ranges from 2 to 5 days, segmented in six-hour segments (thus $|\mathcal{T}| = 12$). For the $i$-th segment, the "ground truth" solar wind $y_i$ is set to its median value over the 6 hours.

DTLR is validated using a nine fold cross-validation (Table 3 in appendix), where each fold is a continuous period corresponding to a solar rotation.[6]

## 6 EMPIRICAL VALIDATION

Table 1 summarizes the DTLR performance on the synthetic and solar wind problems (detailed results are provided in the appendix).

Table 1: DTLR performance: accuracy (MAE and RMSE, the lower the better; Pearson, the higher the better) and stability $\sigma_0$ and $C_1$ (the lower the better). For each indicator, is reported the DTLR value (9-fold CV), the baseline value and the time-lag error.

| Problem | M.A.E | R.M.S.E | Pearson Corr. | $\sigma_0$ | $C_1$ |
|---|---|---|---|---|---|
| **I** | 8.82 / 21.79 / 0.021 | 12.35 / 28.79 / 0.26 | 0.98 / 0.87 / − | 29.8 | 0.14 |
| **II** | 10.15 / 27.40 / 0.4 | 13.70 / 35.11 / 0.67 | 0.95 / 0.73 / 0.70 | 26.83 | 0.16 |
| **III** | 3.17 / 11.01 / 0.17 | 4.63 / 14.99 / 0.42 | 0.98 / 0.79 / 0.84 | 11.84 | 0.09 |
| **IV** | 3.88 / 12.28 / 0.34 | 5.33 / 15.89 / 0.64 | 0.98 /0.79/ 0.81 | 12.18 | 0.13 |
| **Solar Wind** | 56.35 / 66.45 / − | 74.20 / 84.53 / − | 0.6 / 0.41 / − | 76.46 | 0.89 |

### 6.1 SYNTHETIC PROBLEMS

On the easy Problem I, DTLR predicts the correct time lag for $97.93\%$ of the samples. The higher value of $\sigma_0$ in problems I and II compared to the other problems is explained from the higher variance in the effect series $y(t)$.

---

[5] https://omniweb.gsfc.nasa.gov

[6] The Sun completes a rotation (or *Carrington rotation*) in approximately 27 days.

On Problem II, DTLR accurately learns the inverse relationship between $\mathbf{x}_t$, $g(\mathbf{x}_t)$ and $y_t$ on average. The time lag is overestimated in the regions with low time lag (with high velocity), which is blamed on the low sample density in this region, due to the data generation process. Interestingly, Problems III and IV are better handled by DTLR, despite a more complex dynamic time lag relationship. In both latter cases however, the model tends to under-estimate the time lag in the high time lag region and conversely to over-estimate it in the low time lag region.

## 6.2 THE SOLAR WIND PROBLEM

DTLR finds an operational solar wind model (Table 1), though the significantly higher difficulty of the solar wind problem is witnessed by the $C_1$ value close to the degenerate value 1. The detailed comparison with the state of the art Reiss et al. (2019) (Fig. 1, Left) shows that DTLR improves on the current best state of the art (on all variants including ensemble approaches, and noting that median models are notoriously hard to beat). (Fig. 1, Right) shows the good correlation between the predicted solar wind[7] and the measured solar wind.

| Model | M.A.E | R.M.S.E |
|---|---|---|
| WS | 74.09 | 85.27 |
| DCHB | 83.83 | 103.43 |
| WSA | 68.54 | 82.62 |
| Ensemble Median (WS) | 71.52 | 83.36 |
| Ensemble Median (DCHB) | 78.27 | 100.04 |
| Ensemble Median (WSA) | 62.24 | 74.86 |
| Persistence (4 days) | 130.48 | 161.99 |
| Persistence (27 days) | 66.54 | 78.86 |
| Fixed Lag Baseline | 67.33 | 80.39 |
| DTLR | 60.19 | 72.64 |

(a) Comparative assessment on the Solar Wind problem compared to the state of the art Reiss et al. (2019, Table 1)

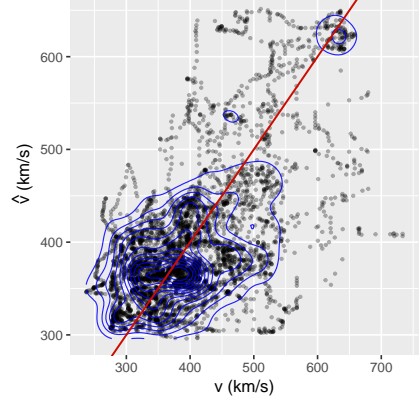

(b) Scatter Chart (9 fold CV)

Figure 1: DTLR on the solar wind problem. Left: comparative quantitative assessment w.r.t. the state of the art (Carrington rotation 2077). Right: qualitative assessment of the prediction.

## 7 DISCUSSION AND PERSPECTIVES

The contribution of the paper is twofold. A new ML setting, Dynamic Time Lag Regression has been defined, aimed at the modelling of varying time-lag dependency between time series. The introduction of this new setting is motivated by an important scientific and practical problem from the domain of space weather, an open problem for over two decades.

Secondly, a Bayesian formalization has been proposed to tackle the DTLR problem, relying on a saddle point optimization process. A closed form analysis of the training procedure stability under simplifying assumptions has been conducted, yielding a practical alternate optimization formulation, implemented in the DTLR algorithm. This algorithm has been successfully validated on synthetic and real-world problems, although some bias toward the mean has been detected in some cases.

On the methodological side, this work opens a short term perspective (handling the bias) and a longer term perspective, extending the proposed nested inference procedure and integrating the model selection step within the inference architecture. The challenge is to provide the algorithm with the means of assessing online the stability and/or the degeneracy of the learning trajectory.

Regarding the motivating solar wind prediction application, a next step consists of enriching the data sources and the description of the cause series $\mathbf{x}_t$, typically by directly using the solar images. Another perspective is to consider other applications of the general DTLR setting, e.g. considering fine-grained modelling of diffusion phenomena.

---

[7]The predicted values, every 6 hours, are interpolated for comparison with the hourly measured solar wind.

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

## APPENDIX A  LOG LIKELIHOOD OF THE LATENT MODEL (3)

### A.1  DIRECT COMPUTATION

Due to the single effect constraint (4) the mixture model (3) can be expressed simply as

$$P(\mathbf{y}|x) = \left( \sum_{i \in \mathcal{T}} \hat{p}_i(x) \prod_{j \in \mathcal{T}} \sqrt{\frac{1 + \alpha_{ji}}{2\pi\sigma^2}} e^{-\frac{1}{2\sigma^2}(1+\alpha_{ji})\left(y_j - \hat{y}_j(x)\right)^2} \right)$$

$$= \left( \sum_{i \in \mathcal{T}} \hat{p}_i(x) \prod_{j \in \mathcal{T}} \sqrt{\frac{1 + \alpha_{ji}}{2\pi\sigma^2}} e^{-\frac{1}{2\sigma^2}\alpha_{ji}\left(y_j - \hat{y}_j(x)\right)^2} \right) \exp\left(-\frac{1}{2\sigma^2}\sum_{j \in \mathcal{T}}\left(y_j - \hat{y}_j(x)\right)^2\right)$$

Let $\theta \overset{\text{def}}{=} (\hat{\mathbf{y}}, \hat{\mathbf{p}}, \sigma, \alpha)$ denote the parameters of the model and consider the probability that predictor $\hat{y}_i$ is the good one conditionally to a pair of observation $(x, \mathbf{y})$:

$$q_i(x, \mathbf{y}) = P(\tau_i = 1 | x, \mathbf{y})$$

$$= \frac{1}{Z(x, \mathbf{y}|\theta)} \hat{p}_i(x) \exp\left(-\frac{1}{2\sigma^2} \sum_{j \in \mathcal{T}} \alpha_{ji}(y_j - \hat{y}_j(x))^2 + \frac{1}{2} \sum_{j \in \mathcal{T}} \log(1 + \alpha_{ji})\right)$$

with

$$Z(x, \mathbf{y}|\theta) = \sum_{i \in \mathcal{T}} \hat{p}_i(x) \exp\left(-\frac{1}{2\sigma^2} \sum_{j \in \mathcal{T}} \alpha_{ji}(y_j - \hat{y}_j(x))^2 + \frac{1}{2} \sum_{j \in \mathcal{T}} \log(1 + \alpha_{ji})\right).$$

This gives immediately

$$\mathcal{L}[\{(x, \mathbf{y})\}_{\text{data}}|\theta] = -|\mathcal{T}| \log(\sigma) - \mathbb{E}_{\text{data}}\left[\sum_{i \in \mathcal{T}} \frac{1}{2\sigma^2} (y_i - \hat{y}_i(x))^2 - \log(Z(x, \mathbf{y}|\theta))\right]$$

## A.2 LARGE DEVIATION ARGUMENT

Even though the log likelihood can be obtained by direct summation, for sake of generality we show how this can result from a large deviation principle. Assume that the number of learning samples tends to infinity, and so that in a small volume $dv = dx d\mathbf{y}$ around a given joint configuration $(x, \mathbf{y})$, the number of data $N_{x,\mathbf{y}}$ becomes large. Restricting the likelihood to this subset of the data yields the following:

$$\mathcal{L}_{x,\mathbf{y}} = \prod_{m=1}^{N_{x,\mathbf{y}}} \sum_{\{\tau^{(m)}\}} \frac{\hat{p}(\tau^{(m)}|x)}{\prod_{i \in \mathcal{T}} \sqrt{2\pi} \, \sigma_i(\tau^{(m)})} \exp\left(-\frac{1}{2} \sum_{i \in \mathcal{T}} \frac{(y_i - \hat{y}_i(x))^2}{\sigma_i(\tau^{(m)})^2}\right).$$

Upon introducing the relative frequencies:

$$q_i(x, \mathbf{y}) = \frac{1}{N_{x,\mathbf{y}}} \sum_{m=1}^{N_{x,\mathbf{y}}} \tau_i^{(m)} \qquad \text{satisfying} \qquad \sum_{i \in \mathcal{T}} q_i(x, \mathbf{y}) = 1,$$

the sum over the $\tau_i^{(m)}$ is replaced by a sum over these new variables, with the summand obeying a large deviation principle

$$\mathcal{L}_{x,\mathbf{y}} \asymp \sum_{\mathbf{q}} \exp\left(-N_{x,\mathbf{y}} \mathcal{F}_{x,\mathbf{y}}[\mathbf{q}]\right)$$

where the rate function reads

$$\mathcal{F}_{x,\mathbf{y}}[\mathbf{q}] = |T| \log(\sigma) + \sum_{i \in \mathcal{T}} \left[(y_i - \hat{y}_i(x))^2 \frac{1 + \sum_{j \in \mathcal{T}} \alpha_{ij} q_j}{2\sigma^2} - \frac{1}{2} q_i \sum_{j \in \mathcal{T}} \log(1 + \alpha_{ji}) + q_i \log \frac{q_i}{\hat{p}_i}\right].$$

Taking the saddle point for $q_i$ yield as a function of $(x, \mathbf{y})$ expression (7). Inserting this into $\mathcal{F}$ and taking the average over the data set yields the log likelihood (5) with opposite sign:

$$\mathcal{L}[\{(x, \mathbf{y})\}_{\text{data}}|\theta] = -\mathbb{E}_{data}\left[\mathcal{F}_{x,\mathbf{y}}[\mathbf{q}(x, \mathbf{y})]\right].$$

## A.3 SADDLE POINT EQUATIONS

Now we turn to the self-consistent equations relating the parameters $\theta$ of the model at a saddle point of the log likelihood function. First, the optimization of the predictors $\hat{\mathbf{y}}$ yields:

$$\frac{\partial \mathcal{L}}{\partial \hat{y}_i(x)} = \frac{1}{\sigma^2} \mathbb{E}_{data}\left[(y_i - \hat{y}_i(x))(1 + \sum_{j \in \mathcal{T}} \alpha_{ij} q_j(x, \mathbf{y})) \Big| x\right].$$

Then the optimization of $\hat{\mathbf{p}}$ gives:

$$\frac{\partial \mathcal{L}}{\partial \hat{p}_i(x)} = \mathbb{E}_{data}\left[\frac{q_i(x, \mathbf{y})}{\hat{p}_i(x)} - \lambda(x) \Big| x\right],$$

$$= \frac{1}{\hat{p}_i(x)} \mathbb{E}_{data}\left[q_i(x, \mathbf{y}) \Big| x\right] - \lambda(x)$$

with $\lambda(x)$ a Lagrange multiplier to insure that $\sum_i \hat{p}_i(x) = 1$ This gives

$$\hat{p}_i(x) = \frac{1}{\lambda(x)} \mathbb{E}_{data} \Big[ q_i(x, \mathbf{y}) \Big| x \Big]$$

Hence

$$\sum_{i \in \mathcal{T}} \hat{p}_i(x) = \frac{1}{\lambda(x)} = 1 \qquad \forall x$$

in order to fulfill the normalization constraint, yielding finally expression (9).

Finally the optimization of $\alpha$ reads:

$$\frac{\partial \mathcal{L}}{\partial \alpha_{ij}} = \frac{1}{2(1 + \alpha_{ij})} \mathbb{E}_{data} \big[ q_j(x, \mathbf{y}) \big] - \frac{1}{2\sigma^2} \mathbb{E}_{data} \big[ \big( y_i - \hat{y}_i(x) \big)^2 q_j(x, \mathbf{y}) \big].$$

## APPENDIX B    PROOF OF PROPOSITION 3.1

Given $I(x)$ a candidate index function we associate the point-like measure

$$p_i(x) = \delta_{i, I(x)}.$$

Written in terms of $p$ the loss function reads

$$\mathcal{L}_2(\hat{y}, p) = \mathbb{E}_{x, \mathbf{y}} \Big[ \sum_{i \in T} p_i(x) \big( y_i - \hat{y}(x) \big)^2 \Big].$$

Under (3) (with $\alpha_{ij} = \alpha \delta_{ij}$) the loss is equal to

$$\mathcal{L}_2(\hat{y}, p) = \mathbb{E}_x \Big[ \sum_{i \in T} p_i(x) \Big( \big( \hat{y}_i(x) - \hat{y}(x) \big)^2 - \hat{p}_i(x) \frac{\alpha \sigma^2}{1 + \alpha} \Big) \Big] + \sigma^2$$

The minimization w.r.t. $\hat{y}$ yields

$$\hat{y}(x) = \sum_{i \in T} p_i(x) \hat{y}_i(x). \tag{16}$$

In turn, as a function of $p_i$ the loss being a convex combination, its minimization yields

$$p_i(x) = \delta_{i, I(x)}, \tag{17}$$

$$I(x) = \arg\min_{i \in T} \Big( \big( \hat{y}_i(x) - \hat{y}(x) \big)^2 - \hat{p}_i(x) \frac{\alpha \sigma^2}{1 + \alpha} \Big). \tag{18}$$

Combining these equations (16,17,18) we get

$$I(x) = \arg\max_{i \in T} \big( \hat{p}_i(x) \big),$$

which concludes the proof.

## APPENDIX C    STABILITY ANALYSIS

The analysis is restricted for simplicity to the case $\alpha_{ij} = \alpha \delta_{ij}$. The log likelihood as a function of $r = \sigma^2 / \sigma_0^2$ and $\beta = \alpha / r$ after inserting the optimal $\mathbf{q} = \mathbf{q}(x, \mathbf{y})$ reads in that case

$$\mathcal{L}(r, \beta) = -\frac{|\mathcal{T}|}{2} \log(r) - \frac{|\mathcal{T}|}{2r} + \frac{1}{2} \log(1 + r\beta) + \mathbb{E}_{data} \Big[ \log(Z) - \lambda(x) \sum_{i \in \mathcal{T}} \hat{p}_i(x) \Big]$$

with

$$Z = \sum_i \hat{p}_i(x) \exp\Big( -\frac{\beta}{2\sigma_0^2} \Delta y_i^2(x) \Big),$$

and where $\lambda(x)$ is a Lagrange multiplier which has been added to impose the normalization of $\hat{\mathbf{p}}$. The gradient reads

$$\frac{\partial \mathcal{L}}{\partial r} = \frac{1}{2r^2}\left(|\mathcal{T}|(1-r) + \frac{\beta r^2}{1+\beta r}\right),$$

$$\frac{\partial \mathcal{L}}{\partial \beta} = \frac{r}{2(1+r\beta)} - \frac{1}{2}C_1[\mathbf{q}],$$

$$\frac{\partial \mathcal{L}}{\partial \hat{y}_i(x)} = \frac{1}{\sigma^2}\mathbb{E}_{data}\Big[\big(y_i - \hat{y}_i(x)\big)\big(1 + \alpha q_i(x, \mathbf{y})\big)\Big|x\Big].$$

$$\frac{\partial \mathcal{L}}{\partial \hat{p}_i(x)} = \frac{\mathbb{E}_{data}\big[q_i(x, \mathbf{y})|x\big]}{\hat{p}_i(x)} - \lambda(x),$$

with

$$C_1[\mathbf{q}] = \frac{1}{\sigma_0^2}\mathbb{E}_{data}\Big(\sum_{i \in \mathcal{T}} q_i(x, \mathbf{y})\Delta y_i^2(x)\Big),$$

This leads to the following relation at the saddle point:

$$r = \frac{|\mathcal{T}| - C_1[\mathbf{q}]}{|\mathcal{T}| - 1},$$

$$\alpha = \frac{|\mathcal{T}|}{|\mathcal{T}| - 1}\frac{1 - C_1[\mathbf{q}]}{C_1[\mathbf{q}]},$$

$$\hat{y}_i(x) = \frac{\mathbb{E}_{data}\Big[y_i\big(1 + \alpha q_i(x, \mathbf{y})\big)\Big|x\Big]}{\mathbb{E}_{data}\Big[1 + \alpha q_i(x, \mathbf{y})\Big|x\Big]}$$

$$\hat{p}_i(x) = \mathbb{E}_{data}\big[q_i(x, \mathbf{y})|x\big].$$

Let us now compute the Hessian. It is easy to see that the block corresponding to the predictors $\hat{\mathbf{y}}$ decouples from the rest as soon as these predictors are centered.

Denoting

$$C_2[\mathbf{q}] = \frac{1}{\sigma_0^4}\mathbb{E}_{data}\Big[\sum_{i \in \mathcal{T}} q_i(x, \mathbf{y})\Big(\Delta y_i^2(x) - \sum_{j=1}^{n} q_j(x, \mathbf{y})\Delta y_j^2(x)\Big)^2\Big],$$

we have

$$\frac{\partial^2 \mathcal{L}}{\partial r^2} = \frac{1}{2r^2}\Big(-|\mathcal{T}| + 2\frac{|\mathcal{T}|}{|\mathcal{T}| - 1}\big(C_1[\mathbf{q}] - 1\big) - \beta^2 C_1^2[\mathbf{q}]\Big)$$

$$\frac{\partial^2 \mathcal{L}}{\partial r \partial \beta} = \frac{1}{2r^2}C_1^2[\mathbf{q}]$$

$$\frac{\partial^2 \mathcal{L}}{\partial \beta^2} = \frac{1}{4}\Big(C_2[\mathbf{q}] - 2C_1^2[\mathbf{q}]\Big)$$

$$\frac{\partial^2 \mathcal{L}}{\partial \hat{p}_i(x) \partial \hat{p}_j(x)} = -\frac{\mathbb{E}_{data}\big[q_i(x, \mathbf{y})q_j(x, \mathbf{y})|x\big]}{\hat{p}_i(x)\hat{p}_j(x)}$$

$$\frac{\partial^2 \mathcal{L}}{\partial r \partial \hat{p}_i(x)} = 0$$

$$\frac{\partial^2 \mathcal{L}}{\partial \beta \partial \hat{p}_i(x)} = -\frac{u_i[x, \mathbf{q}]}{2\hat{p}_i(x)},$$

where

$$u_i[x, \mathbf{q}] \stackrel{\text{def}}{=} \frac{1}{\sigma_0^2} \mathbb{E}_{data} \Big[ q_i(x, \mathbf{y}) \big( \Delta y_i^2(x) - \sum_{j \in \mathcal{T}} q_j(x, \mathbf{y}) \Delta y_j^2(x) \big) | x \Big].$$

There are two blocks in this Hessian, the one corresponding to $r$ and $\beta$ and the one corresponding to derivatives with respect to $\hat{p}_i$. The stability of the first one depends on the sign of $C_2[\mathbf{q}] - 2C_1^2[\mathbf{q}]$ for $|\mathcal{T}|$ large while the second block is always stable as being an average of the exterior product of the vector $(q_1(x, \mathbf{y})/\hat{p}_1(x), \ldots, q_{|\mathcal{T}|}(x, \mathbf{y})/\hat{p}_{|\mathcal{T}|}(x))$ by itself. At the degenerate point $\alpha = 0$, $r = 1$, $\hat{p}_i = 1/|\mathcal{T}|$ the Hessian simplifies as follows. Denote

$$d\eta = dr\mathbf{e}_1 + d\beta\mathbf{e}_2 + \int dx \sum_{i \in \mathcal{T}} d\hat{p}_i(x)\mathbf{e}_{i+2}(x)$$

a given vector of perturbations, decomposed onto a set of unit tangent vectors, $\{\mathbf{e}_1$ and $\mathbf{e}_2\}$ being respectively associated to $r$ and $\beta$, while $\mathbf{e}_i(x)$ associated to $\hat{p}_i(x)$ for all $i \in \mathcal{T}$ and $x \in \mathcal{X}$. Denote

$$\mathbf{u} = \sum_{i \in \mathcal{T}} \int dx u_i[x] \mathbf{e}_i(x)$$

$$\mathbf{v}(x) = \sum_{i \in \mathcal{T}} \mathbf{e}_i(x)$$

with

$$C_2 = \frac{1}{|\mathcal{T}|\sigma_0^4} \mathbb{E}_{data} \Big[ \sum_{i \in \mathcal{T}} \big( \Delta y_i^2(x) - \frac{1}{|\mathcal{T}|} \sum_{j \in \mathcal{T}} \Delta y_j^2(x) \big)^2 \Big].$$

$$u_i[x] = \frac{1}{\sigma_0^2} \mathbb{E}_{data} \big[ \Delta y_i^2(x) - \sigma_0^2 | x \big].$$

With these notations the Hessian reads:

$$H = \frac{1}{2} \Big( -|\mathcal{T}|\mathbf{e}_1\mathbf{e}_1^t + \mathbf{e}_1\mathbf{e}_2^t + \mathbf{e}_2\mathbf{e}_1^t + \big(\frac{C_2}{2} - 1\big)\mathbf{e}_2\mathbf{e}_2^t - \mathbf{u}\mathbf{e}_2^t - \mathbf{e}_2\mathbf{u}^t - \int dx \mathbf{v}(x)\mathbf{v}^t(x) \Big).$$

In fact we are interested in the eigenvalues of $H$ in the subspace of deformations which conserve the norm of $\hat{\mathbf{p}}$, i.e. orthogonal to $\mathbf{v}(x)$, thereby given by

$$\eta = \eta_1\mathbf{e}_1 + \eta_2\mathbf{e}_2 + \eta_3\mathbf{u}.$$

In this subspace the Hessian reads

$$H = \frac{1}{2} \begin{bmatrix} -|\mathcal{T}| & 1 & 0 \\ 1 & \frac{C_2}{2} - 1 & -M|\mathcal{T}|C_2 \\ 0 & -M|\mathcal{T}|C_2 & 0 \end{bmatrix},$$

where $M$ is the number of data points, resulting from the fact that

$$\sum_{i \in \mathcal{T}} \int dx u_i[x]^2 = \frac{M}{\sigma_0^4} \mathbb{E}_{data} \Big[ \sum_{i \in \mathcal{T}} (\Delta y_i^2(x) - \sigma_0^2)^2 \Big],$$

$$= MC_2,$$

because $\mathbb{E}_{data}(\cdot|x)$ as a function of $x$ is actually a point-wise function on the data. If $|u|^2 > 0$ or if $|u| = 0$ and $1 + |\mathcal{T}|(C_2/2 - 1) > 0$ there is at least one positive eigenvalue. Let $\Lambda$ be such an eigenvalue. After eliminating $dr$ and $d\beta$ from the eigenvalue equations in $d\eta$, the deformation along this mode verifies

$$d\eta \propto \Lambda\mathbf{e}_1 + \Lambda(|\mathcal{T}| + \Lambda)\mathbf{e}_2 - M|\mathcal{T}|(|\mathcal{T}| + \Lambda)C_2\mathbf{u},$$

which corresponds to increasing $r$ and $\alpha$ while decreasing for each $x$ the $\hat{p}_i$ having the highest mean relative error $u_i[x]$.

Concerning solutions for which

$$\hat{p}_i(x) = \delta_{i\hat{I}(x)}$$

is concentrated on some index $\hat{I}(x)$, the analysis is more complex. In that case $C_2[\mathbf{p}] = 0$ and $C_1[\mathbf{p}] > 0$. The $(r, \beta)$ sector has 2 negative eigenvalues, while the $\hat{\mathbf{p}}$ block is $(-)$ a covariance matrix, so it has as well negative eigenvalues. The coupling between these two blocks could however in principle generate in some cases some instabilities.

Still, the log likelihood of such solutions reads

$$\mathcal{L} = -\frac{|\mathcal{T}|}{2}\log(\sigma^2) + \frac{1}{2}\log(1+\alpha) - \frac{1}{2\sigma^2}\mathbb{E}_{data}\left[\sum_{i \in \mathcal{T}} \Delta y_i^2(x)\right] - \frac{\alpha}{2\sigma^2}\mathbb{E}_{data}\left[\Delta y_{I(x)}^2(x)\right]$$

so we get the following optimal solution

$$\sigma^2 = \frac{1}{|\mathcal{T}|}\mathbb{E}_{data}\left[\sum_{i \in \mathcal{T}} \Delta y_i^2(x)\right],$$

$$\frac{1}{1+\alpha} = \frac{\mathbb{E}_{data}\left[\Delta y_{I(x)}^2(x)\right]}{\sigma^2},$$

$$I(x) = \arg\min_{i \in \mathcal{T}} \mathbb{E}_{data}\left[\Delta y_i^2(x)|x\right].$$

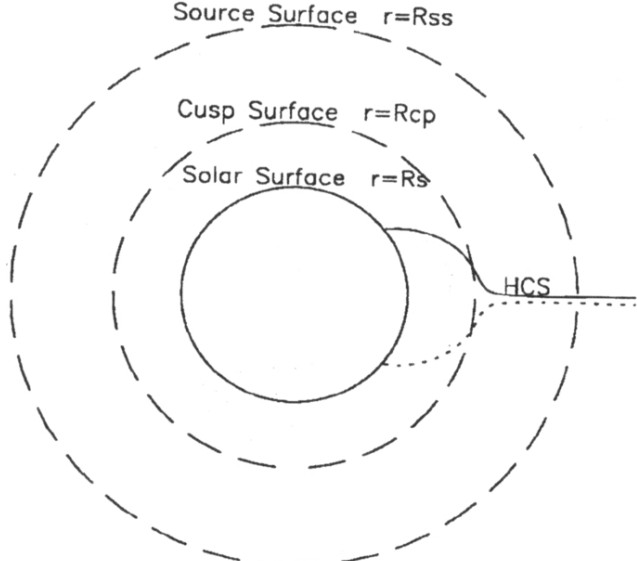

Figure 2: Geometry of the CSSS Model (Zhao and Hoeksema, 1995). The spherical surface with radius $R_{cp}$ is the cusp surface, representing the locus of cusp points of helmet streamers and the location of the inner corona. The source surface (radius $R_{ss}$) is free to be placed anywhere outside the cusp surface. Figure reproduced from Zhao and Hoeksema (1995).

## APPENDIX D    EXPERIMENTS: ADDITIONAL DETAILS

Here we provide some additional details and context to the experimental validation of the DTLR methodology described in section 5. Table 2 provides some information about the datasets used in the synthetic and solar wind prediction problems[8]. Sections D.1.1 and D.1.2 give additional plots for evaluating the experimental results.

For the solar wind prediction task, the solar wind data was mapped into standardized Gaussian space using a quantile-quantile and inverse probit mapping. Nine fold cross-validation was performed using splits as specified in table 3. To compare the DTLR results with the state of the art solar wind forecasting, we used results from Reiss et al. (2019, Table 1). Since Reiss et al. (2019) compared the various forecasting methods on only one solar rotation (first row of table 3), comparing these results with DTLR can be considered as a preliminary examination. Nevertheless, the results presented in table 1a show encouraging signs for the competitiveness and usefulness of the DTLR method.

It is well-established that the solar wind speed observed near the Earth's orbit (at the L1 Lagrange point about a million kilometers upstream solar wind) bears an inverse correlation with the rate (FTE) at which the magnetic flux tubes expand between the photosphere and a reference height called the source surface, typically placed at 2.5 solar radii in many coronal models (Wang and Sheeley, 1990). That is, regions of the Sun with smaller FTE give rise to fast wind and regions with larger values of FTE emanate slow solar wind. This inverse correlation forms the basis of the current solar wind prediction techniques (Poduval and Zhao, 2014; Poduval, 2016) including the WSA-ENLIL model (Wang and Sheeley, 1990; Odstrčil et al., 1996; Odstrčil and Pizzo, 1999a;b; Odstrčil, 2003; Odstrčil et al., 2004) of NOAA/SWPC. Mathematically, FTE is defined as:

$$FTE = \left( \frac{R_\odot}{R_{ss}} \right)^2 \frac{B_r(phot)}{B_r(ss)} \qquad (19)$$

where, $B_r(phot)$ and $B_r(ss)$ are the radial component of magnetic fields at the photosphere and the source surface, and $R_\odot$ and $R_{ss}$ are the respective radii. The coronal magnetic field is computed using CSSS model, the geometry of which is shown in figure 2, by extrapolating the photospheric magnetic field measured by various ground based observatories and spacecraft. For the present work,

---

[8]In the solar wind problem, the training and test data sizes correspond to one cross-validation split

we used the photospheric field measured using the Global Oscillation Network Group (GONG), a network of ground based solar observatories over different places on the globe.

Table 2: Synthetic and Real-World Problems

| Problem | # train | # test | $d$ | $|T|$ |
|---:|---|---|---|---|
| **I** | $10,000$ | $2,000$ | $10$ | $15$ |
| **II** | $10,000$ | $2,000$ | $10$ | $20$ |
| **III** | $10,000$ | $2,000$ | $10$ | $20$ |
| **IV** | $10,000$ | $2,000$ | $10$ | $20$ |
| **Solar Wind** | $77,367$ | $2,205$ | $374$ | $12$ |

Table 3: Cross validation splits used to evaluate  DTLR on the solar wind forecasting task

| Split Id | Carrington Rotation | Start | End |
|---|---|---|---|
| 1 | 2077 | 2008/11/20 07:00:04 | 2008/12/17 14:38:34 |
| 2 | 2090 | 2009/11/09 20:33:43 | 2009/12/07 04:03:59 |
| 3 | 2104 | 2010/11/26 17:32:44 | 2010/12/24 01:15:56 |
| 4 | 2117 | 2011/11/16 07:04:41 | 2011/12/13 14:39:28 |
| 5 | 2130 | 2012/11/04 20:39:43 | 2012/12/02 04:06:23 |
| 6 | 2143 | 2013/10/25 10:17:52 | 2013/11/21 17:36:35 |
| 7 | 2157 | 2014/11/11 07:09:56 | 2014/12/08 14:41:02 |
| 8 | 2171 | 2015/11/28 04:09:27 | 2015/12/25 11:53:33 |
| 9 | 2184 | 2016/11/16 17:41:04 | 2016/12/14 01:16:43 |

## D.1 SUPPLEMENTARY PLOTS

### D.1.1 SYNTHETIC PROBLEMS

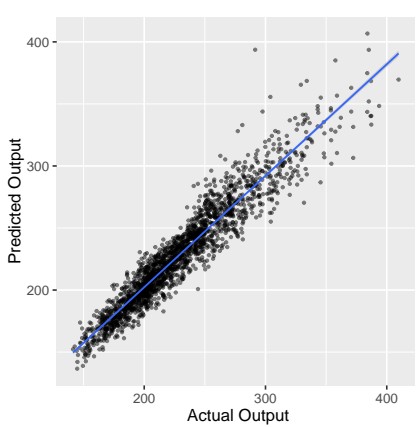

(a) **Problem II**, Goodness of fit, Output $y(x)$

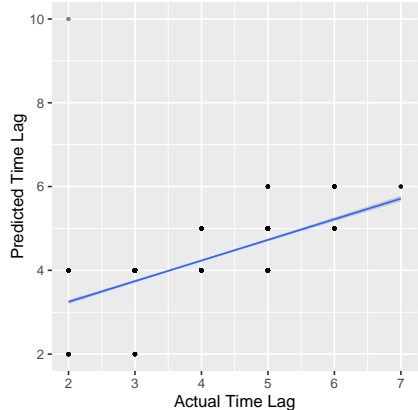

(b) **Problem II**, Goodness of fit, Time lag $\tau(t)$

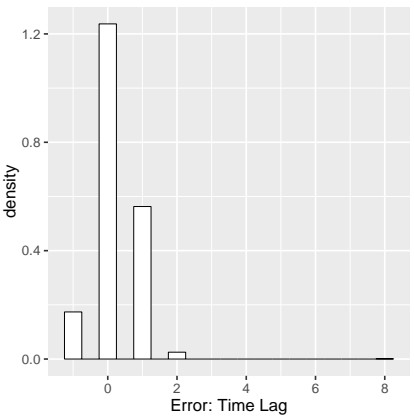

(c) **Problem II**, Error of time lag prediction

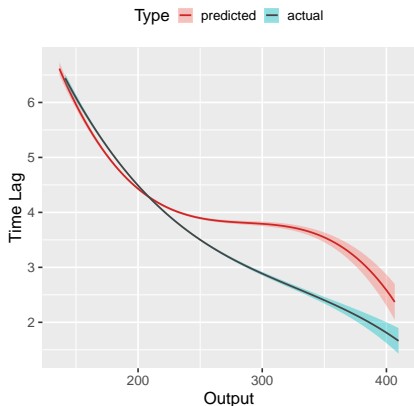

(d) **Problem II**, Output vs Time Lag Relationship

Figure 3: **Problem II**, Results

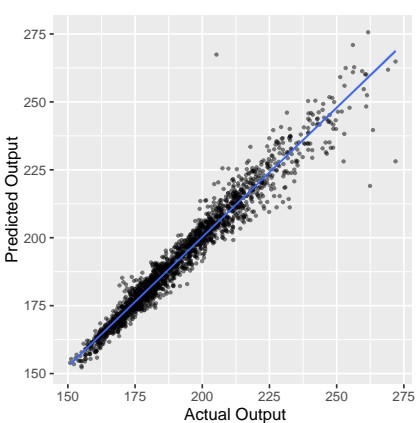

(a) **Problem III**, Goodness of fit, Output $y(x)$

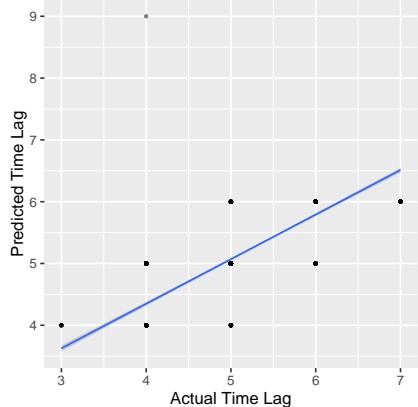

(b) **Problem III**, Goodness of fit, Time lag $\tau(t)$

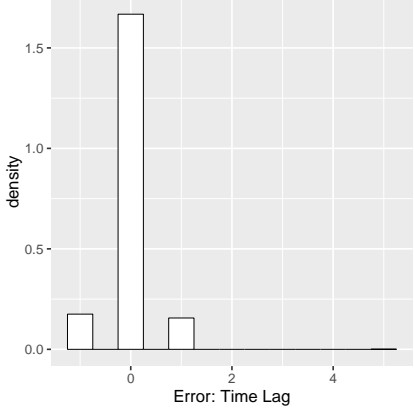

(c) **Problem III**, Error of time lag prediction

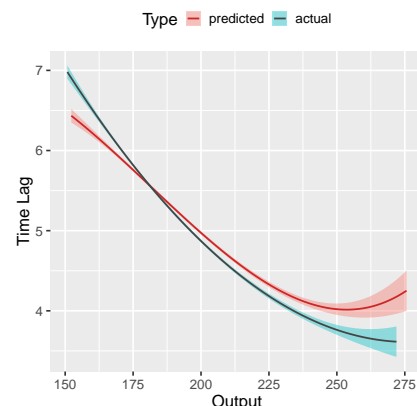

(d) **Problem III**, Output vs Time Lag Relationship

Figure 4: **Problem III**, Results

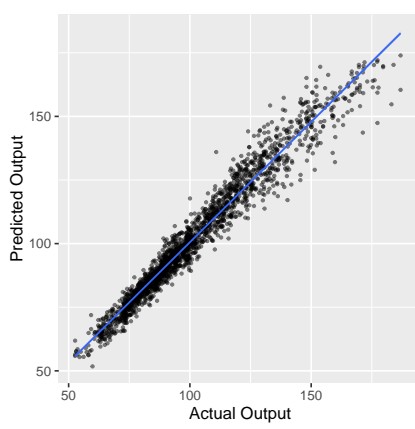

(a) **Problem IV**, Goodness of fit, Output $y(x)$

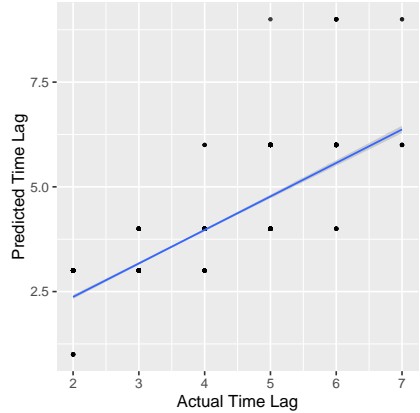

(b) **Problem IV**, Goodness of fit, Time lag $\tau(t)$

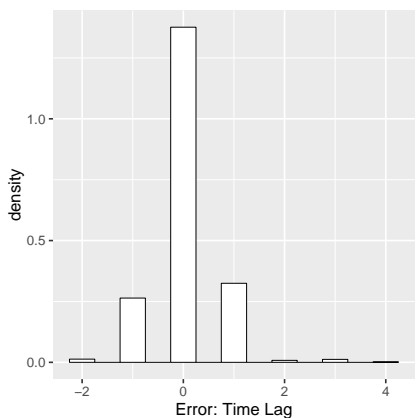

(c) **Problem IV**, Error of time lag prediction

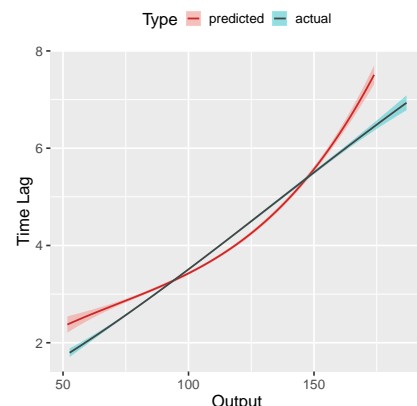

(d) **Problem IV**, Output vs Time Lag Relationship

Figure 5: **Problem IV**, Results

### D.1.2 SOLAR WIND PREDICTION

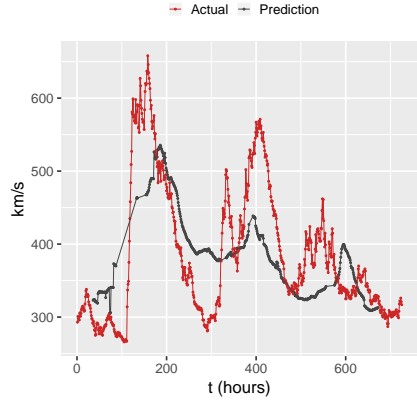

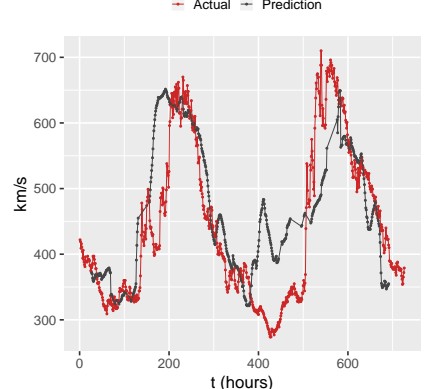

(a) Hourly forecasts for period 2008-11-20 07:00 to 2008-12-17 14:00

(b) Hourly forecasts for period 2016-11-16 17:00 to 2016-12-14 01:00

Figure 6: **Solar Wind Prediction**: reconstructed time series predictions

## APPENDIX E    NEURAL NETWORK ARCHITECTURE DETAILS

Table 4: Network Architecture Details

| Problem | # Hidden layers | Layer sizes | Activations |
|---|---|---|---|
| **I** | 2 | $[40, 40]$ | [ReLU, Sigmoid] |
| **II** | 2 | $[40, 40]$ | [ReLU, Sigmoid] |
| **III** | 2 | $[40, 40]$ | [ReLU, Sigmoid] |
| **IV** | 2 | $[60, 40]$ | [ReLU, Sigmoid] |
| **Solar Wind** | 2 | $[50, 50]$ | [ReLU, Sigmoid] |

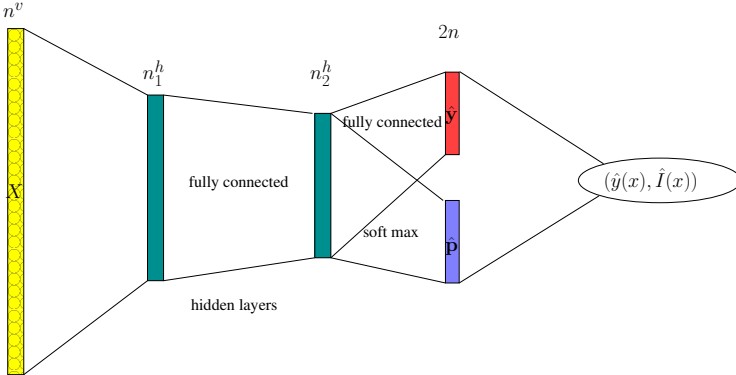

Figure 7: Architecture of the neural network specified by the number of units $(n^v, n_1^h, n_2^h, 2|T|)$ in each layer.

