# OpenReview forum: "Dynamic Time Lag Regression: Predicting What & When"
_ICLR.cc/2020/Conference — Accept (Poster)_

### Official Review · AnonReviewer2 · 2019-10-20
**Official Blind Review #2**

**Rating:** 6

**Review:**

The topic is out of my scope. I talked to the PC about it, and the PC asked me to continue reviewing this paper. So I tried to understand this paper, and I found it is difficult for me. I am not sure if the writing is not good or for some other reasons. I read Section 2.1, and I thought the following description would discuss how to solve \phi(t). Unfortunately, I did not find where \phi(t) is mentioned in Section 2.2. Maybe, I misunderstood Section 2.2.

I do not know how to rate this paper. To avoid killing a good paper, I rated this paper as Weak Accept, and leave the decision to other reviewers and the AC.

**Experience Assessment:**

I do not know much about this area.

**Review Assessment: Checking Correctness Of Derivations And Theory:**

I did not assess the derivations or theory.

**Review Assessment: Checking Correctness Of Experiments:**

I did not assess the experiments.

**Review Assessment: Thoroughness In Paper Reading:**

N/A

---

> ### Author Response · Authors · 2019-11-15
> **Thank you for your time.**
>
> We thank you for your decision of not rejecting the paper!

---

### Official Review · AnonReviewer1 · 2019-10-24
**Official Blind Review #1**

**Rating:** 6

**Review:**

The paper presents a new sequence to sequence regression problem in which every element of the input sequence contributes to the output sequence after an unknown delay. The setup is motivated by a problem in space weather that seeks to predict the solar wind from solar observations.
A mixture of Gaussian-based model is proposed and a closed-form update for the maximum likelihood problem is derived in the paper. After providing extensive analysis for the alternating optimization of the parameters, the proposed model is tested in synthetic and real-world experiments, where it achieves state-of-the-art in solar wind prediction.

The approach of the paper to the problem is principled and it provides extensive theoretical and experimental results for further insight. Unfortunately, one of the main equations (equation 3) is difficult to parse due to its unclear notation and moving forward from that early point, I could not trust my interpretation of the presented model (to be more precise, use of subscript “t” for both x and y on the LHS, the summation index, and the probability space of p all use notations that I found confusing and problematic.)

As a separate note, early on, the paper argues against existing sequences to sequence models for this task due to their limitation in working with individual sentences. However, the paper proceeds to make a modeling assumption of the same sort which bounds the time lag. In light of this, I do think that transformer type models can be adapted to this setting, and in fact, an input dependent attention mechanism is ideal for combining the prediction of time-lag and the raw response in the output sequence. The expectation is for this type of model to perform well in the proposed setting. Could you please comment?


**Experience Assessment:**

I do not know much about this area.

**Review Assessment: Checking Correctness Of Derivations And Theory:**

I did not assess the derivations or theory.

**Review Assessment: Checking Correctness Of Experiments:**

I assessed the sensibility of the experiments.

**Review Assessment: Thoroughness In Paper Reading:**

I made a quick assessment of this paper.

---

> ### Author Response · Authors · 2019-11-15
> **Thank you for assessment, we try to address the issues raised below.**
>
>
> Thank you for pointing that Equation 3 is difficult to read: it has been rewritten.
>
> The definition of the time lag interval $\cal T$ has been clarified, as well as the definition of the sought model ${\bf y}_t$, and we hope that Eq (3) is now clear.
>
>
> Re bounding the time lag/transformer/attention mechanism:
>
> We agree that in principle a transformer or attention mechanism can be used to directly produce the aggregation
> ($\sum{p_{i} \hat{y}_i}  $).
>
> However, DTLR has two merits compared to an attention-based mechanism.
> 1. Algorithmically speaking, an attention-based mechanism would face a significantly lower signal to noise ratio compared to DTLR:
> * in order to train the transformer, the attention-based mechanism must aim to estimate $y(t)$;
> * therefore its input must be ($x(t - i), i \in {\cal T}$) to have the necessary information
> * $x(t)$ is of dimension $D$, circa a few hundred in the space weather application, and the size of $\cal T$ is 12.
> * an attention-based mechanism would thus map a vector of dimension $D \times 12$ onto a scalar;
> * this contrasts with DTLR, the input of which is $x(t)$ (vector of dimension $D$) onto vectors $p$ and $y$, both of dimension 12;
> * DTLR thus faces a signal to noise ratio which is better by an order of magnitude.
>
> 2. Theoretically speaking, DTLR comes with a result of consistency and a stability analysis.
>
> Another issue is that it is not clear how to enforce the constraint "a cause has a single effect" with an attention-based mechanism.

---

### Official Review · AnonReviewer3 · 2019-10-30
**Official Blind Review #3**

**Rating:** 6

**Review:**

The paper is motivated by a specific application about the prediction of solar wind. It is a major threat to the modern world. Authors propose a new regression model called Dynamic Time-Lag Regression which is a special case for spatio-temporal data. For solving this regression problem, the authors use a standard Bayesian approach to fit the model. The analysis of saddle point in Section 3.2 is quite impressive and meaningful. Correspondingly, the practical DTLR algorithm iteratively updates the parameters and hyper-parameters. In experiments, it reveals that the proposed method improves the performance for this particular problem significantly. The application solved by authors is very important, and it is improved by the proposed method. However, the novelty of the work is not presented clearly.

Main argument
1.	It is not clear about the difference between the proposed regression model and others. In my knowledge, it is a quite normal model for spatio-temporal data.
2.	The Bayesian approach and its practical algorithm are very common solutions. What is the highlighted content besides the linear stability analysis?

For the experiments, the following should be addressed.
1.	RMSE in Section 5usually stands for root mean square error. What does relative MSE mean?
2.	Authors said, “After domain knowledge, the time-lag ranges from 2 to 5 days, segmented in six-hour segments.” Does the setting of segmentation have a significant effect?
3.	It would be beneficial to empirically demonstrate the evidence for supporting the theory of avoiding saddle point by synthetic regression.



**Experience Assessment:**

I have read many papers in this area.

**Review Assessment: Checking Correctness Of Derivations And Theory:**

I assessed the sensibility of the derivations and theory.

**Review Assessment: Checking Correctness Of Experiments:**

I assessed the sensibility of the experiments.

**Review Assessment: Thoroughness In Paper Reading:**

I read the paper at least twice and used my best judgement in assessing the paper.

---

> ### Author Response · Authors · 2019-11-15
> **Thank you for the detailed feedback.**
>
> Re novelty (it is a quite normal model for spatio-temporal data):
> In mainstream spatio-temporal problems, there is no dynamic unknown time-lag phenomenon. More precisely, for most problems in biology, ecology, meteorology, medicine and forestry, the time-lag (between the cause series and the effect series) follows a distribution which might be unknown, but is assumed to be stationary.
>
> The Bayesian approach is very common, what is the highlighted content?
> In other problems (e.g. natural language processing and translation), the correspondence between the input/cause series $x(i)$ and the output/effect series $y(i)$ is most often handled using bidirectional approaches or by exploiting the fact that input sentences and output sentences are mapped one-to-one.
> Bidirectional approaches were not considered due to the intended usage of the model: one only sees the sequence up to now in order to predict the future.
> The novelty of the proposed method can be seen by contrast with the vanilla alternative:
> * use the bidirectional approach for the training data,
> * infer independently the lag $\hat{g}(\mathbf{x}(t))$ and the effect $\hat{y}(t + g(\mathbf{x}(t))$
> * use vanilla regression on the two problems $x(t),\ \hat{g}(\mathbf{x}(t))$ and $\mathbf{x}(t), \hat{y}(t + \hat g(\mathbf{x}(t)))$.
> The problem with this vanilla option is that the errors on $\hat{g}$ and $\hat{y}$ will accumulate.
>
> Other issues:
> 1- We use standard RMSE for both the synthetic and real problems, we thank the reviewer for pointing out this typo!
> 2- the segmentation of the 2-5 days ranges into 12 segments; the more the number of segments, the more difficult it is for local models $\hat{y}_i$ to be sufficiently diverse. Experimentally, the approach failed with 60 segments, and we did not investigate further the sensitivity of the approach to the size of $\mathcal{T}$.
> 3- the benefits of the saddle analysis: a clear benefit is to detect when the information in the data is insufficient, and can only lead to the trivial solution (predicting the average). Instead of proceeding by trial and error, we have a characterization of failure cases. In our experiments this is indicated by the value of $C_1$ (see table 1 at the end of the paper). A value significantly smaller than 1 indicates a differentiation of the model between the predictors.

---

### Official Review · AnonReviewer4 · 2019-11-03
**Official Blind Review #4**

**Rating:** 8

**Review:**


This paper proposes a bayesian method, Dynamic Time Lag Regression (DTLR), to predict effect-data y from a cause x.
While both, cause and effect, are time series (x = x(t) and y = y(t)), the cause is assumed to have influence on the effect via a time lag, y(x(t), t) = y(phi(t)) = y(t + g(x(t))). The novelity of the problem posed is that the time lag g(x(t)) is allowed to be non-stationary and supposed to be unknown. The authors outlie a a Bayesian approach. The likelihood of the cause-effect data is conidered used to optimize the regressors for a given, finite set of time lags and the probability of these time lags. This discussion is very brief and detailled derivations are given in the appendix. An accompanying stability analysis is performed. In the implementation a neural network-based approach is used, where in turns the parameters of the model are optimised by a typical gradient descent algorithm and afterwards the hyperparemeters are set to values determined by the parameters. The theoretical deviation is accompanied by practical expirements on synthetically generated data and (real) space weather data. The performance of the model is compared to the one of a prediction with a fixed time lag, which it in fact outperforms. On the space weather (solar wind) data the model is shown to outperform several state-of-the-art models. A codebase for an implementation done in the Scala language is given.

The paper presented provides a meaningful contribution to the field of time series analysis and Bayesian methods. The novelty of the problem imposed as well as the accompanying solution is outlined very clearly. The problem is well justified and the motivation is clearly given.

I recommend to accept it, even though a few details could have been described more precisely.

1. The definition of the time lag g(x(t)) could have been made more clearly, providing a few examples at this point could help to clarify these definitions here.
2. For the synthetic data generation a justification would be helpfull in order to motivate the choice of the functions f and v.
3. A chart with sample time series data (y) against the time would be helpful to better visualize the problem.


**Experience Assessment:**

I have published one or two papers in this area.

**Review Assessment: Checking Correctness Of Derivations And Theory:**

I assessed the sensibility of the derivations and theory.

**Review Assessment: Checking Correctness Of Experiments:**

I assessed the sensibility of the experiments.

**Review Assessment: Thoroughness In Paper Reading:**

I read the paper thoroughly.

---

> ### Author Response · Authors · 2019-11-15
> **Thank you for the kind words !**
>
> 1. The definition of the time lag has been clarified (Eq. 3) and is hopefully more readable.
>
> 2. The synthetic experiments are simplified models of a propagating disturbance, where the disturbance is constant (1), or travels with constant velocity (2) or with constant acceleration (3).
>
> 3. The time lag is illustrated on Fig. 5 in supplementary material, comparing the predicted and observed effect series and showing their chaotic nature. The cause can hardly be represented due to its dimension (circa 500).

---

### Decision · Program_Chairs · 2019-12-19

**Decision:**

Accept (Poster)

**Comment:**

The paper proposes a Bayesian approach for time-series regression when the explanatory time-series influences the response time-series with a time lag. The time lag is unknown and allowed to be non-stationary process. Reviewers have appreciated the significance of the problem and novelty of the proposed method, and also highlighted the importance of the application domain considered by the paper.